# Another Weapon against Cancer and Metastasis: Physical-Activity-Dependent Effects on Adiposity and Adipokines

**DOI:** 10.3390/ijms22042005

**Published:** 2021-02-18

**Authors:** Silvia Perego, Veronica Sansoni, Ewa Ziemann, Giovanni Lombardi

**Affiliations:** 1Laboratory of Experimental Biochemistry and Molecular Biology, Milano, IRCCS Istituto Ortopedico Galeazzi, 20161 Milan, Italy; silvia.perego@grupposandonato.it (S.P.); giovanni.lombardi@grupposandonato.it or; 2Department of Athletics, Strength and Conditioning, Poznań University of Physical Education, 61-871 Poznań, Poland; ziemann.ewa@gmail.com or

**Keywords:** physical activity, adipokines, adiposity, low-grade chronic inflammation, cancer, metastasis, bone metastases

## Abstract

Physically active behavior has been associated with a reduced risk of developing certain types of cancer and improved psychological conditions for patients by reducing anxiety and depression, in turn improving the quality of life of cancer patients. On the other hand, the correlations between inactivity, sedentary behavior, and overweight and obesity with the risk of development and progression of various cancers are well studied, mainly in middle-aged and elderly subjects. In this article, we have revised the evidence on the effects of physical activity on the expression and release of the adipose-tissue-derived mediators of low-grade chronic inflammation, i.e., adipokines, as well as the adipokine-mediated impacts of physical activity on tumor development, growth, and metastasis. Importantly, exercise training may be effective in mitigating the side effects related to anti-cancer treatment, thereby underlining the importance of encouraging cancer patients to engage in moderate-intensity activities. However, the strong need to customize and adapt exercises to a patient’s abilities is apparent. Besides the preventive effects of physically active behavior against the adipokine-stimulated cancer risk, it remains poorly understood how physical activity, through its actions as an adipokine, can actually influence the onset and development of metastases.

## 1. Introduction

Regarding physical activity (PA), lifestyles can be categorized as: (i) inactive; (ii) insufficiently active; (iii) sufficiently active; and (iv) highly active [1,2]. Overall, 2.5 h per week of moderate-intensity activity or 1 h/week of vigorous PA (equivalent to 600 metabolic equivalent task (MET) min/week, with MET measured in units of resting energy expenditure) can be considered the target for the general population to obtain health benefits [3].

Physical inactivity and sedentary behavior represent widespread causes of mortality, and although the clinical attention on this phenomenon has increased in the last two decades, the incidence of adverse events associated with a sedentary lifestyle is growing worldwide. According to the 2009 Global Health Risk report by the World Health Organization (WHO), the world-leading global burdens for mortality are hypertension (responsible for 12.8% of deaths globally), tobacco smoking (8.7%), hyperglycemia (5.8%), physical inactivity (5.5%), and overweight and obesity (4.8%). Other than directly increasing the risk for mortality, these factors are also responsible for increasing the risk of chronic diseases (e.g., heart disease, diabetes, cancers). As a matter of fact, physical inactivity is a leading cause for overweight and obesity, and together they are main cause—and which worsen the effects—of hypertension, hypercholesterolemia, and hyperglycemia, as well as several other risk factors and diseases [4].

A causal link has been established between physical inactivity and overweight and obesity, along with the risk of development and progression of various cancers, mainly in middle-aged and elderly subjects. This association is both direct, as mediated by humoral factors released by tissues and abnormally stimulated in these conditions (e.g., skeletal muscle, adipose tissue, bone), and indirect, as a result of the complex cascade of events generated in all organs and tissues. In total, 21–25% of breast and colon cancer burden, 27% of type 2 diabetes mellitus (T2DM), and about 30% of ischemic heart disease burden have been directly associated with physical inactivity [4]. Further, the association between physical inactivity and obesity increases the risk for tumor recurrences, and among others, colon, breast, and endometrial cancers [5].

Since strict interconnections exist among, although not limited to, a physically active lifestyle, body weight and adiposity control, and cancer prognosis, this review aims to discuss the molecular mechanisms underlying such interconnections. A specific focus is on the role of adipokines, the main mediators of the metabolic inflammation, which are involved in the pathogenesis and progression of cancer and whose circulating levels are strictly dependent upon PA. Articles included in this review have been retrieved from PubMed and Scopus by combing the following terms: “physical activity”, “physical inactivity”, “sedentary behavior”, “physical exercise”, “exercise”, “exercise program”, “training” with “cancer”, “cancer risk”, “transformation”, “metastasis”; and/or with “adipokines”, “myokines”, “adipo-myokines”, “obesity”, “adipose tissue”, “white adipose tissue”.

## 2. Sedentary Behavior, Adiposity, and Low-Grade Chronic Inflammation: Risk Factors for Tumor Development and Metastasis

Sedentary behavior, defined as “any waking behavior characterized by an energy expenditure ≤ 1.5 METs while in a sitting, reclining, or lying posture”, is considered an independent risk factor for cancer and cancer-related deaths. This may differ from physical inactivity, which is defined as “an insufficient physical activity level to meet present physical activity recommendations” (i.e., too little exercise) [6]. Paradoxically, sedentary behavior and PA (even vigorous PA) can coexist, for instance in an office worker who spends long, continuous periods of time sitting but goes for five jogging sessions a week after work, thereby meeting the current PA recommendations, or in a worker with a physically demanding job who spends long periods of time sitting in the evening and on weekends [7].

Sedentary behavior has been associated with the increased risk of several cancers, e.g., colorectal (CRC, 28−44%), breast (BC, 8−17%), and endometrial cancer (28–36%), while association with other cancers is strongly suspected. This is due to the development of metabolic alterations, dysfunctions of sex hormone secretion, and chronic low-grade systemic inflammation (LGI) [7,8,9,10]. LGI relies on increased circulating levels of pro-inflammatory mediators, such as C-reactive protein (CRP), interleukin (IL)-6, and tumor necrosis factor (TNF)α, which generate aberrant inflammatory responses and immune cell activation, in turn leading to the increased production of reactive oxygen species (ROS), with related DNA damage and a reduced DNA repair rate [10]. IL-6, whose key role in LGI is discussed below, induces the hypermethylation of the tumor suppressor gene p53, resulting in the downregulation of pro-apoptotic target genes [11]. Adiposity, which may be associated with body weight and PA status, may act as an intermediate variable linking sedentary behavior to cancer [7]—a positive energy imbalance (i.e., low energy expenditure and high energy intake) results in increased adiposity and weight gain, potentially leading to overweight and obesity. Importantly, obesity is a known risk factor for the same cancers (i.e., BC, CRC, and endometrial cancer) as sedentary behavior, and as with inactivity, an association with other cancers is suspected [12,13]. The mechanisms underlying the association between adiposity and neoplastic transformation are the same as for sedentary behavior, namely metabolic dysfunction, increased levels of sex hormones, LGI, and inflammatory mediators (cytokines and adiposity-related adipokines) [14,15]. For instance, adipose tissue (AT) expresses the aromatases that convert androgens into estrogens, thus increasing the risk for hormone-related female cancers in post-menopausal women. Furthermore, AT-derived adipokines also affect estrogen synthesis [16,17]. AT is capable of sequestering steroid hormones, such as vitamin D, whose decreased bioavailability [18,19] is associated with an increased risk of CRC [20].

AT is an endocrine organ consisting of different cell types (i.e., the stromal–vascular fraction comprising endothelial cells, leukocytes, lymphocytes, and other immune cells) other than the lipid-laden adipocytes, which only account for 20–40% of the cellular content but up to 83% of AT volume (the number of AT stromal–vascular cells outnumbers adipocytes by 3 to 1) [21]. AT can be either white (WAT) or brown (BAT). The former is present in the subcutaneous layer, omentum, and retroperitoneum; stores excess energy in the form of lipids; and is increased in obesity. The latter is mainly present in the humans in the cervical and supraclavicular regions and dissipates energy through thermogenesis [13]. Consequently, white and brown adipocytes express different developmental, morphological, and metabolic features functional to their biological roles. Cell belonging to both fractions (adipocytes and stromal–vascular cells) may have roles in stimulating or supporting tumorigenesis [13]. AT-resident immune cells polarize their phenotype, depending on the adipocytes’ health status [22], towards either a pro-inflammatory type 1 [23], supported by the expression of pro-inflammatory mediators (e.g., TNFα, IL-6, IL-1), or a regulatory type 2, associated with the secretion of anti-inflammatory species (e.g., IL-10, IL-4, transforming growth factor (TGF)-β) [24] and recruitment of CD4+ lymphocytes and Treg cells [25]. Lipid accumulation in WAT, such as in obesity, causes hypoxia and release of chemokines that recruit type-I-polarized (M1) macrophages, CD8+ lymphocytes, and natural killer (NK) cells. A chronic type 1 inflammation may generate a LGI status [26].

In tumor patients, obesity-related AT dysfunctions worsen the prognosis due to increased risk of metastasis [27]. Inflamed AT secretes adipokines and pro-inflammatory cytokines, influencing the tumor microenvironment [28]. Cancers require a favorable environment to progress, as well as adequate energy substrates to satisfy the increased metabolic activity rate. Cytokines and adipokines directly establish an appropriate inflammatory environment that satisfies the tumor’s needs and allows the infiltration of macrophages and lymphocytes, while chemokines and acute-phase proteins participate through the recruitment of immune effectors [22].

In obesity, WAT is enriched in myofibroblasts that deposit a stiff matrix and cause fibrosis (i.e., desmoplasia) [29], a condition that resembles the tumor microenvironment [30]. AT-derived stromal cells (ASCs) and pre-adipocytes are recruited at the tumor site by IL-8 and CXCL1 [31], where they are induced to differentiate into myofibroblasts via TGF-β–mitogen-activated protein kinase (MAPK) signaling. Tumor infiltration by ASCs is associated with a worse prognosis. Myofibroblasts also express vascular endothelial growth factor (VEGF), which stimulates intra-tumor angiogenesis [32].

By inducing hormones and myokines and counteracting adiposity accumulation, PA limits the risk of cancer development and progression. This is better defined for cancer development, while less is known about the effects of PA on tumor progression and metastasis formation.

## 3. The Influence of Physical Activity on Adipokines in the Development and Course of Cancer

Several epidemiological studies have hypothesized and described the relationship between the circulating levels of adipokines and cancer morbidity [33]. In this section, we describe the role of adipokines in tumorigenesis and tumor development.

### 3.1. Adiponectin

Adiponectin (ACRP30), a peptide secreted by AT (although not exclusively), is involved in glucose metabolism and energy homeostasis [34]. Its actions resemble those of exercise, since it increases glucose uptake by skeletal muscles (SKM) and suppresses hepatic gluconeogenesis [35]. Among the adipokines, it is the only one whose circulating levels are inversely related to fat mass, and in fact obesity is characterized by low plasma levels of adiponectin and decreased expression of adiponectin receptors (AdipoR1, AdipoR2) [36].

Adiponectin has pro-apoptotic and anti-proliferative properties [37]. Decreased adiponectin concentrations are found in subjects affected by different cancers (i.e., BC, CRC, endometrial, esophageal, and liver) [38]. Treating BC cell lines (i.e., MCF-7, MDA-MB-231, T47D) with adiponectin increases the apoptosis rate and inhibits progression throughout the cell cycle [39,40,41]. Adiponectin inhibits adhesion, invasion, and migration of BC cells by activating the adenosine-monophosphate-dependent kinase (AMPK)/S6K axis and the consequent upregulation of the tumor suppressor gene LKB1 [42]. AMPK activation inhibits several signaling pathways, such as MAPK, phosphatidylinositol-3-kinase (PI3K)/Akt, WNT-β-catenin, nuclear factor κB (NF-κB), and JAK2/STAT3 [43]. These effects depend upon the estrogen receptor (ER) status of BC cells. In ER cells, adiponectin suppresses cell growth, proliferation, invasion, and migration and induces apoptosis; in ER+ cells, adiponectin at low concentrations allows the interaction of the adaptor protein APPL1 with AdipoR1, ERα, insulin-like growth factor (IGF1R), and c-Src. These complexes activate MAPK signaling and consequently promote the proliferation of BC cells [44]. Adiponectin can also be found within adipocyte-derived exosomes [45,46]. Adiponectin and AdipoR1/R2 downregulation has been associated with an increased degree of malignancy in endometrial carcinoma [47,48]. Further, in human and mouse CRC cell lines, combined treatment with adiponectin and metformin, a glycemia-lowering drug used in T2DM, reverses the cancer-inducing effect of IL-1β [49], similarly to non-small-cell lung carcinoma (NSCLC) cells, in which adiponectin inhibits migratory and invasive capacities and induces the expression of epithelial markers while downregulating the mesenchymal markers [50].

### 3.2. Leptin

Leptin is a peptide hormone secreted by AT in amounts related to fat mass. It regulates food intake and energy expenditure by acting in the central nervous system (CNS), SKM, liver, and AT itself [51] through the leptin receptor (LEPR) [52], whose activation affects multiple intracellular pathways, including JAK/STAT3, MAPK, PI3K/Akt, extracellular-signaling-regulated kinase (ERK)1/2, AMPK, and insulin receptor substrate (IRS) [33]. Leptin resistance, i.e., the inability of leptin to exert its biological effects due to non-functioning or underexpressed receptors, as observed in obesity, leads to deregulated cytokine signaling and increased appetite and energy consumption, in turn further stimulating inflammation and AT accumulation [53]. Epidemiological studies have associated high circulating leptin levels with an increased risk of developing tumors such as BC and CRC [54]. In addition, in vitro, leptin stimulates the proliferation of breast, colon, prostate, pancreatic, ovarian, and lung cancer cells [55].

The oncogenic action of leptin in breast tissue involves the JAK/STAT3 and PI3K pathways [56,57], induction of anti-apoptotic genes, VEGF-dependent stimulation of angiogenesis [58], and enhancement of estrogen signaling (e.g., aromatase upregulation, ERα activation, p53 suppression) [59]. In BC patients, high circulating leptin levels increase the risk of tumor progression, since its possible involvement in the autocrine loop that sustains the expression of an inflammatory phenotype [60].

High levels of leptin released from excessive AT may drive the progression of gastrointestinal cancers. In gastric cancer (GC), leptin expression is associated with that of the proto-oncogene human epidermal growth factor receptor (HER)2, which in turn is associated with invasion, lymph node metastasis, tumor stage [61] and expression of VEGF [61], intercellular adhesion molecule (ICAM)-1 (CD54) [62], and matrix metalloproteinase (MMP)14, which plays crucial roles in tumor invasion [63]. Similarly, in human colon cancer cells (HCT116, SW620, SW480), leptin stimulates migration through the activation of JAK/STAT3 [64].

Other studies suggest a correlation between circulating leptin and pancreatic cancer. LEPR overexpression in pancreatic tumor tissue stimulates proliferation, migration, and angiogenesis, while reducing apoptosis rates. Furthermore, in PANC-1 tumor cells, leptin induces MMP13 expression and stimulates migration and invasion [65]. Leptin overexpression promotes human pancreatic cancer xenograft growth and lymph node metastasis in mice, inducing the progression throughout the cell cycle and the expression of pancreatic cancer stem cell (CSC) markers (CD24+, CD44+, ESA+, ALDH+) [66].

### 3.3. Apelin

Apelin is a widely expressed peptide (e.g., in the brain, lungs, kidneys, pancreas, testicles, prostate, AT) that acts through the apelin receptor (APJ), an angiotensin II type-1 receptor with whom it is often co-expressed, and mediates several effects, such as reduction of blood pressure, apoptosis, and angiogenesis; promotion of cell proliferation; and regulation of glucose homeostasis [67,68,69,70]. Apelin is significantly induced in various tumors and has a role in cell proliferation (especially on CSCs), tumor development and metastasis (by stimulating angiogenesis), and drug resistance [71]. Apelin stimulates MCF-7 BC cell proliferation via the dose-dependent induction of cyclin D1 and amplified in breast cancer 1 (AIB1) and invasion potential by upregulating MMP1 expression [72], a mechanism also described in A549 lung adenocarcinoma cells [73]. Apelin is upregulated in NSCLC compared to healthy lung tissue and is associated with tumor growth and progression [74]. In BC and lung cancers, the combined inhibition of apelin and angiogenesis (i.e., sunitinib) powerfully reduces tumor growth and angiogenesis rates. Various studies have demonstrated that apelin is a key regulator of differentiation, proliferation, and survival of mesenchymal stem cells (MSCs), which in turn regulate the population of CSCs throughout the secreted cytokines and chemokines [75,76,77].

### 3.4. Visfatin

Visfatin is encoded by the nicotinamide phosphoribosyltransferase (NAMPT) gene and is firstly recognized as a cytokine-like pro-differentiating agent of immature B-cell precursors (pre-B-cell colony-enhancing factor (PBEF)) [78]. The NAMPT gene encodes for a 52 kDa protein that can be detected both intracellularly (iNAMPT) and extracellularly (eNAMPT/visfatin). Furthermore, iNAMPT catalyzes the reaction of nicotinamide with 5-phosphoribosyl-1-pyrophosphate to yield nicotinamide mononucleotide (www.genecards.com (accessed on 14 February 2021)), while the secreted form acts as a pro-inflammatory cytokine, namely visfatin [79]. Visfatin is produced by adipocytes, infiltrating macrophages, SKM, the liver, immune cells, cardiomyocytes, and the brain [80], and its circulating levels correlate with BMI and are associated with insulin resistance [81]. Higher visfatin levels are found in both plasma and tumor tissues of patients with different types of cancer compared to their healthy counterparts [82]. In thyroid cancer (TC) patients, visfatin levels are upregulated in cases of metastasis [83], such as in NSCLC. In vitro in NSCLC cell lines (A549 and H358), 48 h of treatment with 100 ng/mL visfatin enhances cell migration and invasion in the Boyden chamber via upregulation and activation of MMP2 and MMP9 [84]. Visfatin modulates several pathways, including ERK1/2, p38-MAPK, and PI3K/Akt. Further, NF-κB inhibition abolishes visfatin-induced cell migration and MMP2/9 upregulation [84]. Visfatin time- and dose-dependently sustains the epithelial–mesenchymal transition (EMT), a process needed to allow the entrance of tumor cells into the bloodstream and their migration [85], and requires the loss of an epithelial marker (E-cadherin) and the upregulation of a mesenchymal marker (*N*-cadherin) in U2OS osteosarcoma cells via NF-κB [86].

### 3.5. Resistin

Resistin is encoded by the RTN gene and is characterized, as are the other members of the protein family, by a C-terminal 10-cysteine residue (putative binding site) [87]. Resistin was discovered in mice treated with the anti-diabetic thiazolidinedione as a factor downregulated in mature adipocytes. Further, resistin was induced and secreted during adipocyte differentiation and its serum levels were inversely related to feeding status [88]. In humans, resistin is mainly expressed by peripheral blood mononuclear cells (PBMCs), macrophages (i.e., those infiltrating AT), and bone marrow cells and plays a role in inflammation and ROS production, as in obesity-associated LGI. Resistin binds to different receptors, including toll-like receptor 4 (TLR4), an isoform of decorin, receptor tyrosine kinase-like orphan receptor 1 (ROR1), and adenylyl cyclase-associated protein 1 (CAP1) [89]. In monocytes, it binds to CAP1 and activates and upregulates NF-κB [90]. Resistin plays a role in tumor growth; in PC-3 and DU-145 PC cells, it activates the PI3K/AKT pathway and induces cell proliferation [91]. Resistin levels are higher in sera of BC patients; treatment of MDA-MB-231 BC cells and MC7 fibroadenomatous cells stimulates cell migration, as evaluated by scratch testing [92]. The binding to TLR4 stimulates PI3K and MAPK (p38), which both activate NF-κB, and consequently stimulates expression of proinflammatory mediators, adhesion molecules (ICAM and vascular cell adhesion molecule (VCAM)), and stromal cell-derived factor (SDF)-1, a transcription factor important in cancer development and metastasis [33].

### 3.6. Ghrelin

The gastric peptide ghrelin regulates energy homeostasis and stimulates growth hormone (GH) secretion, release of acidic species from the gastric mucosa, insulin secretion, and gastric–intestinal mucosa turnover [93,94,95,96] throughout the binding with two splice variants of the G protein-coupled GH secretagogue receptor GHSR, type 1a (GHSR1a) and 1b (GHSR1b) [97]. It is involved in development and progression of several cancers—it stimulates proliferation of CRC cells via the GHSR/Ras/PI3K/Akt/mTOR axis [98], GLUT-1-dependent proliferation of oral cancers [99], GHSR/NF-κB- and GHSR/PI3K/Akt-dependent migration and invasion of gastric [100] and pancreatic adenocarcinoma [101], and ERK2-dependent angiogenesis [102]. Differing from GHSR1a, whose expression in tumors has not been demonstrated [103], GHSR1b has been detected in BC cell lines, and at high amounts in malignant breast tissue, but not in normal mammary tissue [104]. The ghrelin–GHSR axis has been associated with metastasis, as it stimulates the production of nitric oxide (NO) and enhances the phosphorylation of NO synthase (NOS) [105]. In pancreatic adenocarcinoma, blockage of the GHSR/PI3K/Akt pathway has demonstrated the role played by ghrelin in promoting cell migration and invasion [101].

### 3.7. Chemerin

The retinoic acid receptor responder 2 (RARRES2), also known as chemerin, is a ubiquitously expressed adipokine that is relatively abundant in AT and the liver and is involved in inflammatory responses. It acts throughout the activation of two receptors, chemokine-like receptor 1 (CMKLR1) and G-protein-coupled receptor 1 (GPR1) [106]. A third receptor, C-C motif chemokine-like receptor 2 (CCRL2), seems to be involved in local effects [107]. Chemerin is a chemoattractant with either anti- or pro-inflammatory properties, depending on the environmental context. Its concentrations in the blood are increased in obesity due to the increased expression in hypertrophic adipocytes [108], and may contribute to adiposity-related dyslipidemia, LGI, hypertension, and insulin resistance [109]. Chemerin plays either a protective or promoting role in cancer, depending on the context [110]. To escape immunosurveillance, some tumors inhibit chemerin expression by hypermethylating the gene; further, chemerin recruits anti-tumor immune cells [109,111]. In BC, it may act as a tumor suppressor [112] by recruiting immune effector cells [113] when it is expressed at higher levels in the tumor than in the adjacent healthy tissues [114], while IL-1β, TNFα, IL-6, and interferon (IFN)γ upregulate CCRL2 [115].

Chemerin inhibits the development of hepatocellular carcinoma (HCC) metastasis via the upregulation of tumor suppressor phosphatase and tensin homolog (PTEN) and by interfering with the PTEN–CMKLR1 interaction. This results in decreased Akt phosphorylation and suppressed migration, invasion, and metastatic potential of HCC cells in vitro [116]. In vivo, chemerin decreases p38 MAPK and β-catenin phosphorylation in adrenocortical cancer cell xenografts in mice, thus acting as a tumor suppressor [117]. In contrast, in GC cells it enhances p38 and ERK1/2 MAPK phosphorylation, leading to increased invasiveness, as well as upregulation of VEGF and MMP7 [118]. The chemerin-mediated induction of MMPs have also been reported in esophageal squamous cell carcinoma (ESCC) and neuroblastoma [119,120].

### 3.8. Lipocalin 2

Lipocalin (LCN)2, a member of the lipocalic protein family that transports small lipophilic ligands, was first discovered in neutrophils, forming complexes with MMP9 to prevent its self-degradation and to increase its activity in vitro (also referred to as neutrophil gelatinase-associated lipocalin (NGAL)) [121,122]. The role of MMP9 in the degradation of ECM and the basement membrane, NGAL and the MMP9/NGAL complex is thought to contribute to tumor progression, invasion, and metastasis [123]. LCN2 is involved in inflammation [124] and kidney and liver damage [125,126,127], and is upregulated in various epithelial carcinomas, including breast, ovarian, lung, colon, and pancreas carcinomas [128,129,130,131,132]. LCN2 overexpression in BC is associated with EMT (upregulation of mesenchymal vimentin and fibronectin and downregulation of epithelial E-cadherin), motility, and invasion [133]. Macrophage-derived LCN2 induces EMT in MCF-7 cells, promotes local invasion of the ECM, and stimulates migration [134]. It also induces the expression of the pro-angiogenic mediators hypoxia-inducible factor-1α (HIF-1α) and VEGF in MCF-7 and MDA-MB-436 BC cells [135]. In ESCC, NGAL upregulation stimulates migration, invasion, and lung metastasis, while Raf/MEK/ERK pathway activation enhances MMP9 activity [136]. NGAL has been detected in 77% of lung cancers, while its suppression results in E-cadherin induction and vimentin, MMP9, and MMP2 downregulation [137]. LCN2 is involved in TNF-related apoptosis-inducing ligand (TRAIL)-mediated apoptosis in CRC. The mRNA expression data for 71 CRC tissues have revealed that the expression of TRAIL-R2 is inversely related to LCN2 and LCN2 knockdown in CRC cells, which increases the sensitivity to TRAIL via p38 MAPK/CHOP-induced upregulation of TRAIL-R2 [138].

### 3.9. Osteopontin

OPN is a glycoprotein secreted by osteoblasts, osteocytes, and hematopoietic cells (neutrophils, dendritic cells (DCs), NK cells, T and B cells [139]). It is involved in mineralization and bone modeling and remodeling by serving as an anchor to ECM for osteoclasts [140]; it stimulates MSC differentiation towards the osteoblast instead of the adipocyte lineage [141]. It also regulates innate and adaptive immunity [142]. In obesity, OPN plasma levels and expression in AT macrophages increase, while they are reduced by weight loss [143,144]. OPN expression is upregulated in tumor cells in vivo and in vitro, such as in the breasts, stomach, lungs, ovaries, and melanomas [145,146,147,148,149]. OPN plays crucial roles in EMT, in wound healing and in metastasis [150]. In BC cells, OPN induces HIF-1α and EMT-related transcription factors via the PI3k/AKT pathway [151,152]. In HCC, OPN stimulates cancer growth and metastasis by activating (PI3K)/Akt, MAPK, NF-κB, and MMP2 [153]. In NSCLC tissue, VEGF and OPN are overexpressed in association with tumor progression [154]. OPN-dependent migration involves integrins and CD44 and can be inhibited by anti-OPN antibodies [155], while its binding to αvβ3 integrin activates multiple signaling cascades (i.e., PKCα/c-Src/IKK/NF-κB, which increases CD44; MMP9; cyclooxygenase (COX)2 expression; prostaglandin E (PGE)2 release) involved in prostatic tumorigenesis [156]. OPN is induced in pancreatic cancer cells in vitro and in patients’ sera [157], while OPN inhibition decreases VEGF and MMP9 levels, which are involved in pancreatic tumor development and metastasis [158].

### 3.10. IL-6 and TNF-α

Cytokines produced in WAT during chronic inflammation, as in obesity, can promote tumor growth via NF-κB activation [159]. IL-6 concentrations are substantially increased in the tumor microenvironment, as it has pro-tumorigenic effects, increasing survival, growth, angiogenesis, and invasion. IL-6 binds to IL6R receptor and its co-receptor glycoprotein (gp)130, forming the hexamer signaling complex, consisting of two IL-6 molecules, two IL-6R, and two gp130, which can activate several signaling pathways: namely JAK/STAT, MAPK, PI3K, and Src/YAP. A soluble form of IL-6R (sIL-6R), originating from ADAM10/17 (a disintegrin and metalloproteinase-domain-containing protein)-mediated cleavage, or more rarely from alternative splicing, may induce trans-signaling by binding gp130 on cell surfaces [160]. IL-6 has both anti- and pro-inflammatory properties, depending on the inflammatory milieu; the classical signaling mediated by IL-6R exerts an anti-inflammatory action, while the trans-signaling mediated by sIL-6R exerts pro-inflammatory actions [161]. IL-6 is upregulated in several cancers (breast, colorectal, ovarian, lung, and pancreas cancers) in both the tumor tissue and in the patient’s serum and is often associated with advanced disease and poor prognosis [160].

TNFα is mainly produced by macrophages and its biologically active form is a 17 kDa compound that that acts via two receptors: TNFR1 and TNFR2. Both receptors are expressed in adipocytes. Soluble TNFα is derived from transmembrane TNFα (24 kDa), which is cleaved by ADAM17, also known as TNFα-converting enzyme (TACE) [162]. As with L-6, TNFα is expressed at high levels in tumors, together with elevated circulating levels that correlate with advanced disease and poor prognosis. TNFα exerts its action through NF-κB activation and promotes cell proliferation, angiogenesis, and metastasis and inhibits apoptosis [163].

### 3.11. Adipokines and Bone Metastases

Bone and bone marrow are “favorable environments” for circulating cancer cells to metastasize. In fact, about 70–80% of cases of BC and PC display bone involvement [164,165]. The tight and complicated cross-talk among the different cell types residing in bone may provide a supportive niche for cancer cells. Cancer cells, in turn, can express and secrete parathyroid-hormone-related peptide (PTHrP), which induces RANKL expression in osteoblasts and downregulates osteoprotegerin (OPG), hence stimulating proliferation and differentiation of osteoclast precursors and activating mature osteoclasts. Bone resorption causes the release of several factors as well as calcium, which stimulate proliferation of cancer cells and further expression of PTHrP. This vicious circle of events translates into formation of osteolytic lesions and progression of metastases [166,167,168].

As with BAT and WAT, the marrow AT (mAT) is also endocrinally active, however unlike the other forms is less related to fat mass; mAT responds to GH, insulin, and thyroid hormones by releasing fatty acids (FAs) and inflammatory mediators [169,170]. They above make available to cancer cells a lipid-based energy source that sustains proliferation, migration, and invasion [171]. Furthermore, mAT adipokines are key regulators in bone metastasis [172]. Leptin, for instance, promotes migration of BC cells to the bone marrow niche, as well as oncogenesis and proliferation of cancer cells within the niche [173]. Contrarily, adiponectin has anti-cancer effects [33,174,175], and similarly to leptin, its expression in mAT is greater than in WAT [176], especially in cancer patients.

IL-6 is also upregulated in the tumor microenvironment, where it activates two signaling pathways: (i) JAK2/STAT-3, which induces EMT (upregulation of E- and N-cadherin, MMP7, and MMP9) and increases the metastatic potential [177]; (ii) PI3K/Akt, which promotes cancer cell survival [178]. In general, all cytokines and chemokines secreted by the adipocytes, including IL-1α, IL-1β, IL-8, IL-15, IL-16, CXCL1, and CXCL2, have roles in bone metastases [179]. Upregulation of IL-1β and leptin has been involved in BC cell recruitment in mAT [173]. IL-6, TNFα, CXCL12, and leptin promote cell proliferation, migration, and resistance to chemotherapy in multiple myeloma (MM) [180]. In PC, CXCL1 and CXCL2 stimulate osteoclastogenesis, thus promoting the progression of associated bone disease [181]. Additionally, mAT plays a key role in making the bony microenvironment hospitable for cancer cells through the deregulated release of adipokines and cytokines.

Table 1 summarizes the main pro-tumorigenic and pro-metastatic effects of key adipokines.

## 4. Effects of Physical Activity on Adipose Tissue, Low-Grade Inflammation, and Tumor Progression

Regular exercise and a physically active lifestyle are non-pharmacological interventions that prevent overweight and obesity [184], which are often proposed in association with a balanced diet [185]. PA counteracts adiposity-related deregulation of energy balance by stimulating lipolysis (i.e., release of free FA (fFA)) and fFA oxidation (mainly, but not only, in SKM) and improves adipocyte size and adipokine secretion [186,187]. In this sense, endurance training (ET) is regarded as the most effective strategy [188].

In WAT, exercise induces mitochondriogenesis and the expression of brown adipocyte-specific genes, leading to the phenotypic shift towards “beige” or “bright” AT, while systemically ameliorating the high-fat diet (HFD)-induced glucose intolerance [187,189]. Beige adipocytes can be found within WAT; they appear as morphologically similar to brown adipocytes, as they contain multilocular lipid droplets and several mitochondria and express the uncoupling protein (UCP)1, which enhances mitochondrial respiration and non-shivering thermogenesis [186]. UCP in WAT can be induced by the SKM-derived myokine irisin, a cleavage product of membrane FNDC5 induced by muscle contraction [190]. In general, regular exercise decreases the secretion of pro-inflammatory mediators from AT and SKM, such as TNFα, leptin, and MCP-1, and improves LGI status in obesity [187]. Rodent studies have demonstrated that different exercises (swimming, treadmill running, voluntary exercise) over different timeframes (11 days to 8 weeks) improve the mitochondrial activity within WAT [191,192]. Impaired insulin-induced Akt phosphorylation in inguinal fat in mice after 7 weeks on HFD was recovered after acute exercise (2 h running on a motorized treadmill at 5% tilt), which also induced IL-6 and IL-10 expression and decreased M1 macrophage infiltration [193]. Insulin-stimulated Akt phosphorylation and GLUT4 expression in SKM has been induced in the soleus muscles of diabetic mice who underwent high-intensity interval training (HIIT) more than in mice subjected to moderate-intensity training, indicating the greater effectiveness of HIIT in improving glucose metabolism [194].

High levels of mAT, primary fat depot found with a HFD, correlate with low bone mass; voluntary wheel running exercise decreased mAT deposits in both normal and HFD groups and increased trabecular bone volume, cortical bone area, and periosteal and endo-cortical perimeters in C57BL/6 female mice [195]. Similarly, voluntary wheel running exercise in HFD-induced obese mice made mAT adipocytes smaller and induced the pro-lipolytic perilipin3 expression in bone tissue [196]. Further, HFD-fed C57BL/6 mice experienced AT inflammation; exercise on a motorized running treadmill reduced M1 CD11c+ macrophages and CD8+ T-cell, IL-6, and TNFα expression [197]. In obese mice, resistance exercise (RE) stimulated the release of meteorin-like (Metrnl) from SKM, a myokine that improves energy expenditure and heat dissipation in WAT and glucose tolerance; pushes the M2 phenotype shift in macrophages; and induces the regulatory cytokines IL-10, IL-4, and IL-13 [198]. After a 10 week EE program, HFD was associated with increased glycaemia and inflammatory markers compared to a normal diet; fat pads and inflammatory markers (e.g., TNFα, HIF-1α, VEGF-A) were reduced by the combination of EE and whey protein consumption [199].

Human studies have investigated the impacts of exercise on the inflammatory profile of AT. However, it must be considered that obese subjects have limited physical ability compared to healthy persons, and therefore can sustain exercise regimens of reduced intensity and duration. A 12 week supervised, progressive, combined aerobic and RE program undertaken by 21 obese men enhanced oxygen consumption and muscle strength and improved insulin sensitivity, as shown in abdominal subcutaneous AT biopsies. Nevertheless, the expression of markers of mitochondria biogenesis and function, AT browning, lipolysis, as well as inflammatory cytokines and adipokines, was not affected [200]. In otherwise healthy overweight subjects, a 6 week HIIT program increased both mitochondrial content and oxidative phosphorylation in SKM but not in AT; however, WAT oxidative capacity was not improved. Subjects also recorded their dietary habits in the week preceding the start of training and during the training—energy intake, macronutrient composition, and protein intake remained unchanged [201]. A mild exercise intervention over 6 months in healthy sedentary men significantly increased the expression of genes involved in oxidative phosphorylation in the subcutaneous WAT, whereas the Wnt and MAPK signaling pathways were downregulated [202]. The real “browning or beiging” effect of exercise on subcutaneous WAT depots associated with heat production is somehow contradictory in humans [203]. Unlike in rodents, human subcutaneous AT is probably in continuum with dermal AT, and human BAT is actually a mixture of white, beige, and brown adipocytes [204]. Moreover, human UCP1 shares less than 80% homology with the murine one, suggesting the existence of inter-species functional differences [205]. A 3 week training protocol with lean and overweight sedentary individuals had had no impact on both mitochondrial (mt)DNA amount or expression of oxidative and thermogenic genes in WAT [206]. Sedentary adults who underwent 12 weeks of cycling training experienced improvements in several metabolic parameters, among them insulin sensitivity and blood levels of adiponectin, apelin, and irisin (particularly in the overweight group), with no effect on body composition, congruent with subcutaneous WAT browning [207].

Given its modulatory roles in LGI and tumors development, several studies underline the beneficial effects of exercise on adipokines and the overall inflammatory profile [208,209]. Aerobic exercise interventions of various durations (4 weeks to 1 year) have been shown to decrease serum leptin concentrations, together with fat mass [210,211]. In particular, long-lasting protocols, in association with increasing energy expenditure, are more effective in decreasing the concentration of leptin in healthy individuals, while the effects are more pronounced in obese and metabolic syndrome patients who experience heavy exercise-dependent leptin level modulation [212]. The effects of PA on adiponectin levels are more controversial—while mild-to-moderate exercise in healthy, lean subjects has no effect, long-lasting bouts effectively induce mRNA expression in SKM. In obese individuals, sub-maximal aerobic exercise (45 min workout on cycle ergometer at an intensity corresponding to 65% of VO_2_max) only improves insulin sensitivity. Contrarily, acute aerobic exercise and short-term training (3 bouts of treadmill running at either low (50% VO_2_max) or high (75% VO_2_max) intensity) have been shown to increase plasma adiponectin [208]. Chronically elevated secretion of IL-6 from AT (i.e., IL-6 as an adipokine)—as occurs in overweight, obesity, and T2DM–adversely affects insulin sensitivity and glucose metabolism. Instead, when secreted by the contracting SKM (i.e., myokine), it enhances blood glucose uptake by SKM and AT and stimulates lipolysis, thereby mediating the beneficial effects of PA. In general, IL-6 increases following moderate-to-severe exercise in untrained, normal-weight individuals, while decreases in response to strenuous activity. In obese individuals, IL-6 expression in AT is unchanged or reduced, depending on the training modality [213]. PA also protects against TNFα-induced insulin resistance by reducing its circulating levels and increasing the release of the anti-inflammatory agents IL-4 and IL-10. Untrained women with an average BMI of 31 kg/m^2^ underwent a 12 week brisk walking or slow jogging on treadmills 3 days/week and experienced decreased serum IL-1β, IL-6, TNFα, and IFN-γ and increased IL-10. Participants were encouraged to maintain their dietary habits and no differences were recorded in terms of caloric intake or macronutrient composition [214]. Similarly, 4 weeks of moderate intensity walking, together with the maintenance of the usual dietary regimen, resulted in decreased TNFα levels in overweight and obese participants, without any alteration of body weight and composition or levels of adiponectin and CRP, thus suggesting that PA may impair the basal inflammation independently from changes in fat mass [215].

In obesity, the increasing hypoxic pressure enhances oxidative stress and expression of angiogenic and inflammatory mediators, which push up LGI. AT-derived ASCs infiltrate cancer lesions and promote the establishment of a pro-tumor, pro-metastatic microenvironment via paracrine and cell-to-cell contact routes. Further, in obesity, the recruitment of AT cells to tumors is enhanced. These alterations represent further proof of the link between obesity and increased risk of cancer and mortality [33]. By favorably unbalancing the metabolism, PA reduces AT volume and modulates adipokine secretion, immune functions, and the inflammatory state, thereby reducing the risk of carcinogenesis and increasing survival in persons who have had cancer [216] (Figure 1).

Sedentary lifestyle induces an increasing infiltration of macrophages and immune system cells in the WAT. The increased secretion of adipokines stimulates tumor growth and migration and results in a favorable microenvironment for metastasis engraftment. On the contrary, a physically active status and regular exercise reduce the secretion of adipokines, thereby limiting their support of tumor growth and metastasis.

Individuals respond differently to exercise interventions, depending on their subjective fitness (i.e., the maximum capacity for physical effort that an individual can sustain), which in turns depends upon gender, nutrition, and genetics. The genetic background is a substantial contributor to the adaptation to exercise [217]. For instance, family members similarly respond to training, supporting the hypothesis that it is half heritable and half acquired based on the environmental stimuli [218].

In vitro and in vivo studies have analyzed the effects of various training protocols on tumor development and progression. In a microfluidic system mimicking the blood pulsing circulation, intense PA-like shear stresses (60 dynes/cm^2^, 4 h) comparable to those generated within the femoral artery kill 90% of cancer cells (MDA-MB-231, UACC-893, A549, 2008) via necrosis, during circulation, and apoptosis, within 24 h post-circulation [219]. Mechanical stimulation of osteocytes (1 Pa, 1 Hz) reduces MDA-MB-231 BC cell extravasation [220]. Sera from 20 BC women who underwent a combined training protocol (30 min warm-up, 1 h RE exercise, 30 min high-intensity spinning) decreased MCF-7 and MDA-MB-231 cell viability by 11%-19% and tumorigenesis by 50% when inoculated in mice [221]. Blood and platelet-rich plasma from 30 sedentary healthy men who underwent either a high- (HIE) or mild-intensity exercise (MIE) protocol enhanced pro-thrombotic and anti-MMP activity in platelet–nasopharyngeal carcinoma cell line (NPC) interactions (HIE) and reduced thrombosis risk (MIE) when subjected to shear stress (0 and 5% dyne/cm^2^) [222].

In rats with 1-methyl-1-nitrosourea-induced mammary tumors, 35 weeks of moderate-intensity training (5 days/week, 60 min/day) reduced tumor mass and histological and malignant lesions with no lung metastasis as compared to untrained rats [223]. Similarly, voluntary running limited hyperlipidemia, enhancer of BC growth, and metastasis onset in ApoE^−/−^ hyperlipidemic mice with orthotropic BC [224]. In contrast, metastasis spread was not affected in mice with spontaneous Lewis lung carcinoma and experimental (B16BL/6) metastases who underwent voluntary running 9 weeks before cancer cell injection and 2 weeks post-injection of B16BL/6 and after Lewis carcinoma removal [225]. This is in agreement with a 12-study systematic review that highlighted neither positive nor negative effects of PA on metastasis formation, probably due to model heterogeneity [226]. However, exercise has a positive effect on cancer in terms of volume reduction and tumor growth [227].

Clinical studies have evidenced positive effects of PA on both tumor progression and metastasis spread, especially in BC patients. The revision of 700 unique exercise interventions supported the safety and psychophysical effectiveness of exercise in individuals with cancer; exercise importantly delayed disease progression, improved survival, and reduced chemotherapy-associated toxicity [228]. Of 60 patients with stable bone metastases undergoing radiation therapy, in those subjected to 30 min a day of RE training (3 times/week) started the first day of radiotherapy, bone metastases did not increase, while they increased by 16.7% in the untrained group, despite having the same survival rates at 12 and 24 months [229]. In another study, 20 stage IIB–IIIC BC patients underwent 20–45 min cycle ergometer sessions (55–100% VO_2_max, 3 times/week, 12 weeks) and experienced decreased tumor hypoxia; enhanced tumor vascularization; increased circulating endothelial progenitor cells; and decreased IL-1β, IL-2, and NF-κB signaling [230]. In 23 stage I–III CRC patients randomized to either low- (150 min/week) or high-dose (300-min/week) 6-month aerobic activity at 50–70% of the age-predicted maximum HR, circulating tumor cells was significantly reduced, along with BMI, insulin, and soluble ICAM-1 [231]. Among 1354 PC patients, 79% of those who self-reported vigorous PA behavior at least 1/week in the year preceding diagnosis showed reduced metastasis risk compared to less trained subjects. In these subjects, PA modified the methylation profile of the CRACR2A gene promoter encoding for a Ca^2+^-binding protein involved in immunity [232].

Diet also has important effects on metabolic profile and LGI, and indeed, the dietary approach enters into the therapeutic path in cancer treatment. Furthermore, bioactive compounds in diet may directly affect both LGI and tumorigenesis [233,234,235]. However, according to a recent review, most of the studies on PA interventions in cancer patients do not properly consider nutritional aspects [236].

## 5. How the PA-Dependent Effects on Adipokine and Low-Grade Inflammation Affect the Risk of Developing Metastases

Adipokines act as both hormone-like and cytokine-like mediators on target cells [237]. Other than being important in sustaining malignant transformation and cancer cell survival, these mediators have a theoretically pivotal role across the process underpinning metastasis—from EMT to intravasation, survival in circulation, extravasation, seeding, and colonization [85,238]. Given the above-described roles of adipokines in tumor biology and given the well-known effects of exercise on adipokines, in this section we will attempt to identify the molecular effects exerted by PA on the spread of metastases. To date, few studies have addressed this topic.

The pancreatic cancer cells PANC1 and AsPC1 express Ob-Rb (a leptin receptor). Treatment with leptin increases their migration capacity, as evaluated by scratch testing, via activation of Jack2/Stat3 and induction of MMP13. Stably overexpressing leptin PANC1 induces more lymph node metastasis than wild-type cells in mice. In metastasis patients, Ob-Rb expression is increased together with MMP13 levels [65]. Circulating leptin affects pancreatic carcinogenesis by promoting angiogenesis, proliferation, migration, and invasion [239]. Osteocytes exposed in vitro to an oscillatory fluid flow mimicking loading during mild exercise expressed low levels of IL-6, enhanced MDA-MB-231 BC cell migration, and limited osteoclast differentiation and migration, while increasing their apoptosis rate [240].

IL-6 overexpression has been observed in many types of cancer [160]. In mice, 6 weeks of spontaneous PA before liposarcoma injection and 8 weeks of voluntary wheel running post-tumor injection increased IL-6, FABP4, PPAR-γ, autophagy markers, and tumor growth [241]. In C57BL/6 male mice orthotopically transplanted with transgenic mouse PC1 cells, 8 weeks of voluntary access to a wheel reduced the expression of pro-metastatic genes and plasma levels of the angiogenic cytokines IL-6 and CXCL-1, while increasing MEK/MAPK and PI3K/mTOR signaling, HIF-1α, and VEGF [242]. Similarly, 9 weeks of regular moderate swimming (8 min/day) limited liver cancer growth and lung metastasis onset, thus prolonging survival. Contrarily, overload swimming (16–32 min/day) stimulated tumor growth and lung metastasis. Interestingly, moderate swimming suppressed TGF-β1-induced EMT of transplanted liver cancer cells in another study [243].

Bone metastases are the most frequent secondarism. They cause severe pain, pathological fractures, and drastic worsening of life quality. Forty-two stage II or III CRC patients who underwent resection plus adjuvant treatment participated in a training protocol consisting of 50 min home-based exercise 3 times/week (10 min warm stretching, 30 min circular combined aerobic and anaerobic workout, and 10 min wrap-up stretching). Muscle-to-fat ratios were recorded in the untrained group but not in the exercise group; circulating leptin increased in the untrained group, while adiponectin increased in the trained group [244]. Twenty-four sessions (twice a week, 12 weeks) of 75 min of training (10 min warm-up, 2 × 25 min moderate-intensity aerobic exercises at 70% VO_2_max and muscle strengthening, 10 min stretching) combined with a balanced diet improved the BMI, metabolic profile, and insulin resistance in 42 overweight and obese BC survivors, along with decreasing leptin (not significant) and adiponectin (significant) [245]. Stage I–III BC obese women, after the completion of the radiotherapy or chemotherapy cycles, underwent 16 weeks of combined EE and RE, including 150 min/week of moderate–to-vigorous aerobic activity and 2–3 days/week RE. The training group experienced decreased plasma CRP, leptin, IL-6, and IL-8 and increased adiponectin. The AT biopsies displayed reduced amounts of infiltrating M1 macrophages and increased M2 macrophages compared to controls and to baseline, together with a reduction of ex vivo secretion of IL-6 and TNFα and increased secretion of adiponectin and IL-12 [246].

Table 2 summarizes the studies on the effects of PA interventions on cancer development and progression.

## 6. Practical Recommendations and Conclusive Remarks

In the real world, prescribing an exercise program can be difficult due to several factors. First, exercise prescription should be tailor-made for the person’s specific features, with particular attention to comorbidities and age. Second, exercise prescription implies a careful screening of the subject, with the appropriate recording of their anamnestic history together with an accurate physical examination aimed at defining the person’s abilities and capabilities in performing PA, possible health risks, and adverse events, and also determining activity goals. Third, a prescription implies a consultation about the most appropriate times for advancing and reducing the activity (e.g., adaptation to the prescription), especially for those who have never engaged in physical exercise, as well as maintenance dosing. Fourth, the way prescriptions are written must be standardized; for instance, the frequency, intensity, time, and type (FITT) or frequency, intensity, time, type, volume, and progression (FITT-VP) methods are considered the easiest ways [3]. However, all of these major issues, together with several other minor issues, require specific competencies that are not always appropriate for the prescribing specialist. Therefore, on one hand there is a need to assist physicians and patients with adequately and specifically trained personnel (e.g., physiotherapists, sport scientists), while on the other hand there is a need to appropriately train physicians in order to advise them about the information resulting from such consultations. Indeed, it should be kept in mind that exercise is a therapy or a part of a therapeutic path, and indeed can be considered a polypill [248].

In 2010, the American College of Sports Medicine (ACSM) provided guidelines for PA in cancer survivors. According to these general recommendations, inactivity should be avoided and exercise should be carried out continuously and as often as possible [249]. However, the ACSM also provided tumor-specific recommendations for both aerobic and resistance exercises based on the known comorbidities associated with specific cancer types (e.g., increased risk for pathologic fractures from bone metastasis or lymphedema secondary to mastectomy). Indeed, contrary to the common school of thought, resistance and strengthen training are safe and useful in patients with secondary lymphedema [250]. The recommended approach is “low and slow” in order to avoid overtraining, which may have adverse effects, such as in hematopoietic stem cell transplantation in subjects who have had hematologic malignancies because of potential adverse immune effects [249]. Current ACSM recommendations for exercise in subjects with dysmetabolic conditions are 150 min/week of moderate-intensity PA. However, given the dose–effect, 250–300 min/week of moderate-intensity PA may improve weight loss and better prevent weight regain [251,252].

Exercise prescription for the different phases of disease in tumor patients is still debated. There is universal agreement about the need for personalization and progression of exercise, as well as for targeting specific symptoms. Any type of activity is considered beneficial for most patients with a low fitness level. The American Cancer Society (ACS) and ACSM advise against inactivity and suggest moderate regular exercise during and after treatment [249,253]. During treatment, the aim is to stay as fit as possible but the intensity of exercise may be reduced, and usually sedentary patients must start with short sessions of low-intensity exercises [254]. On the other hand, a paper published by De Backer and co-workers demonstrated the beneficial effects (increase maximal oxygen capacity and amelioration of exercise-induced fatigue) of HIIT in 37 patients after radiotherapy and chemotherapy with diverse types of cancers. Regardless, analysis of workload details revealed that this type of exercise relies mainly on aerobic intervals [255]. Patients under treatment, even if hospitalized and bedridden, can benefit from exercise. Patients undergoing chemo- or radiotherapy should avoid exercising when white blood cells < 0.5 × 10^9^/L, hemoglobin< 6 mmol/L, platelets< 20 × 10^9^/L, and temperature> 38°C. Moreover, patients with bone metastases should not lift heavy weights when strength training [254]. In case of disease resolution or stabilization, exercise should be guided by blood cell counts. Moreover, the following advice might be considered—avoid exercise for anemic subjects and avoid public gyms in cases of low leukocyte counts or when an immunosuppression treatment is ongoing, as well as in cases of abnormal levels of minerals in the blood (e.g., sodium, potassium), unresolved pain, or nausea and vomiting [253]. Chlorine in swimming pools may exacerbate skin irritation in cases of radiation treatment. Finally, fatigue and poor fitness are severe adverse events related to cancer and anti-cancer treatments, while exercise, both during and after treatment, may improve these factors [256].

In conclusion, PA contributes substantially to energy expenditure, and therefore to the regulation of energy balance and fat mass. Hence, PA may contribute to metabolic health through beneficial changes within AT. Indeed, exercise is an established therapeutic strategy to treat overweight and obesity and the associated inflammation, and given the intimate association with LGI and cancer, it can be a valid aid for the treatment of tumors [257]. The literature suggests that PA may mitigate cancer-treatment-related side effects and also highlights the importance of encouraging cancer patients to engage in moderate PA several times a week; however, the need to customize and adapt exercises to the patient’s ability is apparent. Despite the mechanistic effects of PA on the regulation of key proteins in metastases, little is understood about how and how much PA, as an adipokine modulator, could actually influence the development of metastases. A deep knowledge of the physiology of adipokines (e.g., in response to exercise) may help in understanding their role in the pathogenesis of cancer. This deepened knowledge may lead to direct use or inhibition of adipokines, somehow mimicking a response to exercise, in order to limit the development of the disease. Among other factors, adiponectin is promising, and its therapeutic “use” [258] could be an area for future research. Finally, a neglected but important aspect to be considered is the history of PA rather than the actual PA status. Limited (or absent) PA in children and adolescents, together with malnutrition statuses (either hypo- or hyper-nutrition), as described in emerging countries, may predispose individuals to the risk of developing cancer, and eventually to more aggressive forms in the future [259].

## Figures and Tables

**Figure 1 ijms-22-02005-f001:**
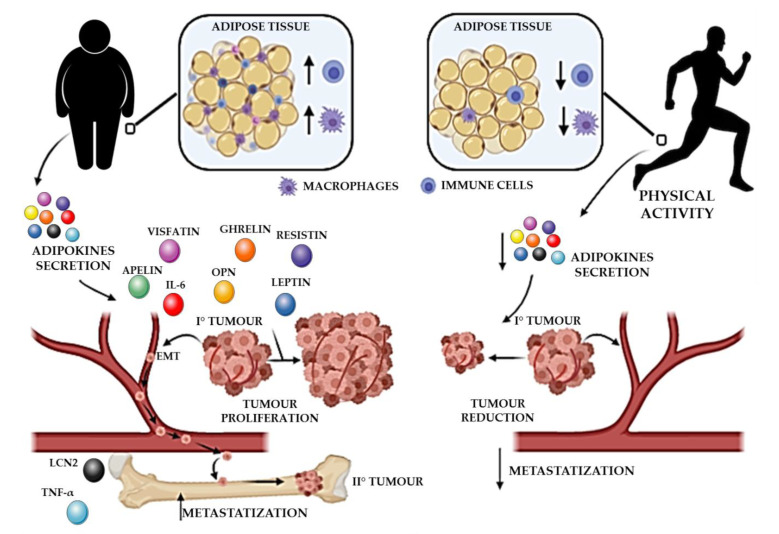
Roles of adipokines in tumor progression—sedentary lifestyle vs. physical activity.

**Table 1 ijms-22-02005-t001:** The roles of adipokines in cancer progression.

	Tumor Progression and Metastases	Ref.
**Adiponectin**	• Inhibits proliferation, adhesion, invasion, and migration of BC cells • Inhibits proliferation of CRC cells by blocking the cell cycle • In NSCLC positively modulate the EMT markers • In endometrial carcinoma reduction of AdipoR1 and AdipoR2 increase malignancy	[42,44] [182] [50] [47]
**Leptin**	• Transform breast epithelial cells to mediate further proliferation of tumor cells • Induce angiogenesis by stimulating the production of VEGF • Acts via the JAK/STAT3 and PI3K pathways • High levels, in GC, correlates with invasion, metastasis and VEGF expression • Significantly improved the migratory activity of human CRC cells • Over-expression of LEPR induces proliferation, migration and angiogenesis in pancreatic cancer cells	[60] [60] [56] [61] [64] [65]
**Visfatin**	• In NSCLC enhanced cell migration and invasion via activation of MMP2 and MMP9 • In osteosarcoma cell lines induces the EMT transition via NF-κB pathway	[84] [86]
**Resistin**	• In prostate cancer cells induce cell proliferation through PI3K/AKT pathway • Promote the invasiveness of BC cells • Activating the signalling of PI3K and MAPK improves tumor growth and metastasis	[91] [92] [33]
**Apelin**	• In BC and lung adenocarcinoma cancer cells promote the invasion of malignant cells via upregulation of MMP1 expression • Induce VSMCs migration through the PI3K/Akt/FoxO3a signalling and by upregulating MMP2 • In NSCLC its higher expression correlates with tumor growth and progression	[72,73] [183] [74]
**Ghrelin**	• Increase angiogenesis via ERK2 signalling pathway • In CRC cell induces proliferation through the GHS-R/Ras/PI3K/Akt/mTOR axis • Promote oral cancer proliferation via modulation of GLUT1 expression • Promote migration and invasion in GC and pancreatic adenocarcinoma	[102] [98] [99] [100,101]
**Chemerin**	• In BC exerts a tumor suppressor role by binding to its CMKLR1 and GPR1 receptors • Suppresses HCC migration, invasion and metastasis through upregulation of tumor suppressor PTEN • In GC cells increase invasiveness and upregulation of VEGF and MMP7	[112] [116] [118]
**Lipocalin2**	• In BC cells induce EMT, and increase motility and invasion • Induce production of HIF-1α and VEGF to stimulate angiogenesis • In ESCC its overexpression promote migration, invasion and lung metastasis and increased MMP9 activity	[133] [135] [136]
**Osteopontin**	• In BC cells increase in EMT-related transcription factors and in HIF-1α and promote angiogenesis, skeletal metastasis and enhancing tumor progression • In HCC facilitates cancer growth and metastases by activation of (PI3K)/Akt, MAPK, NF-kB and MMP2 pathways. • In NSCLC induces tumor cell migration acting on integrins and CD4437	[151,152] [153] [155]
**IL-6**	• Through activation of Jak2/STAT-3 signalling induce EMT, upregulate MMP7 and MMP9, and IL-6 • Activates the PI3K/Akt pathway and promote the survival of cancer cells	[177] [178]
**TNFα**	• Through NF-κB activation promotes cell proliferation, apoptosis, angiogenesis and metastasis	[163]

AdipoR1/R2: Adiponectin Receptor 1/2; BC: Breast Cancer; CMKLR1: Chemokine-Like Receptor 1; CRC: Colorectal Cancer; EMT: Epithelial–Mesenchymal Transition; ESCC: Esophageal Squamous Cell Carcinoma; GC: Gastric Cancer; GPR-1: G-Protein-Coupled Receptor 1; HCC: Hepatocellular Carcinoma; HIF-1α: Hypoxia-Inducible Factor-1α; LEPR: Leptin Receptor; MMP: Matrix Metalloproteinase; NF-κB: Nuclear Factor κB; NSCLC: Non-Small-Cell Lung Carcinoma; PTEN: Phosphatase and Tensin Homolog; VEGF: Vascular Endothelial Growth Factor; VSMC: Vascular Smooth Muscle Cells.

**Table 2 ijms-22-02005-t002:** Effects of exercise training on tumor progression.

In Vitro Studies	Model of Exercise		Effects on Tumor Progression	Ref.
Human MDA-MB-231 and UACC-893 BC cells, A549 lung cancer cells, 2008 ovarian cancer cells	Microfluidic circulatory system: low shear stress of 15 dynes/cm^2^ (resting state) vs. high shear stress of 60 dynes/cm^2^ (intensive exercise)		- 4 h of high shear stress produce necrosis in 90% of circulating cancer cells derived from multiple types of tumor cells - 16–24 h of high shear stress cause apoptosis via anoikis in 92% of circulating cancer cells (vs. 11% of low shear stress) - 8 h of high shear stress destroyed 74% of metastatic BC cells	[219]
Osteocytes-like cells, MDA-MB-231 BC cells, RAW264.7 osteoclasts, and HUVECs	Osteocyte-like cells subjected to 2 h of oscillatory fluid flow with peak shear stress of 1 Pa (mild exercise)		- CM from flow-stimulated osteocyte-like cells increase by 45% migration of BC cells - CM from osteoclast cultured in CM from flow-stimulated osteocytes reduced BC cell migration by 47% - CM from HUVEC cultured in CM from flow-stimulated osteocytes increased BC cell apoptosis by 29% → anti-metastatic potential of flow-stimulated osteocytes mediated by osteoclasts and endothelial cells	[240]
Osteocyte-like cells, MDA-MB-231 BC cells, and HUVECs	3D microfluidic tissue model with osteocytes mechanically stimulated by a physiological oscillatory fluid flow with a peak shear stress of 1 Pa		↓ trans-endothelial BC cell extravasation (32.4% with mechanically stimulated osteocytes vs. 53.5% static osteocytes)	[220]
**In Vivo Studies**	**Groups**	**Exercise Intervention**	**Effects on Tumor Progression**	**Ref.**
Rats with BC induced by carcinogen MNU	-sedentary MNU (*n* = 15) -sedentary CTRL (*n* = 10) -EX MNU (*n* = 15) -EX CTRL (*n* = 10)	35 week-moderate exercise training on a treadmill 60 min/day, 5 days/week	EX MNU developed less tumors per animal than sedentary MNU (2.30 vs. 2.55). sedentary MNU showed pulmonary nodules No metastasis in EX MNU	[223]
Mice inoculated with liposarcoma (LIPO)	-EX CTRL (*n* = 9) -EX LIPO (*n* = 9) -CTRL (*n* = 9) -LIPO (*n* = 9)	-6 weeks spontaneous wheel PA before tumor injection (*n* = 36) -8 weeks voluntary wheel running post-tumor injection (*n* = 18)	↑ IL-6 circulating levels in EX LIPO vs. LIPO and EX CTRL vs. CTRL ↑ tumor growth in EX LIPO vs. LIPO ↓ body weight loss in EX LIPO vs. all remaining groups ↑ risk of developing lung metastasis in EX LIPO vs. LIPO	[241]
C57BL/6 male mice inoculated with mouse prostate adenocarcinoma cells	-EX group (*n* = 28) -CTRL group (*n* = 31)	8 weeks voluntary access to a wheel 24 h/day	↓ pro-metastatic genes in EX group vs. CTRL ↓ IL-6 and CXCL-1 in EX group vs. CTRL ↑ HIF-1α and VEGF (tumor vascularization) in EX group vs. CTRL	[242]
C57BL/6 mice transplanted with murine liver cancer cells	-CTRL (*n* = 12) -8min/d (*n* = 12) -16min/d (*n* = 12) -32min/d (*n* = 12)	-9 weeks of regular moderate swimming, 8 min/day -9 weeks of overload swimming, 16 and 32 min/day	↓ tumor volume and lung metastasis in 8 min/d vs. CTRL ↓ TGF-β1-induced EMT in 8 min/d vs. CTRL ↑ tumor growth and lung metastasis in both 16 min/d 32 min/d vs. CTRL	[243]
Hyperlipidaemia ApoE^-/-^ mice with orthotopic murine BC	-CTRL (*n* = 10)-HEx high-exercise (*n* = 10)-LEx low-exercise (*n* = 10)	-HEx group: 10 km/day of continuous wheel access -LEx group: 8 km/day of wheel access every 2nd day	No differences between HEx, LEx, and CTRL in tumor growth ↓ internal metastases and tumor hypoxia in both Hex and LEx vs. CTRL ↑ beneficial changes in tumor microenvironment	[224]
Mice injected with metastasis of Lewis lung carcinoma (LLC) or with metastasis of B16BL/6 melanoma	-subcutaneous injection of spontaneous LLC cells (*n* = 30)-intravenous injection of experimental B16BL/6 cells (*n* = 30)-CTRL (*n* = 30)	-9 weeks of voluntary running before cancer cells injection -2 weeks of voluntary running after metastases injection (B16BL/6) or surgical removal of the primary tumor (LLC)	no differences in the numbers or sizes of lung metastases between B16BL/6 or LLC groups ↓ plasma insulin and leptin levels in LLC vs. CTRL ↑ adiponectin levels and PDGF-BB in LLC vs. CTRL no effect on VEGF and MCP-1 levels	[225]
Balb/C mice injected with mouse BC cells	-CTRL (*n* = 20) -EX group (*n* = 16)	4 weeks of voluntary wheel running after cancer injection	↑ secondary metastases nodules in lungs in EX group vs. CTRL ↓ endothelial function in EX group vs. CTRL	[247]
**Clinical Studies**	**Groups**	**Exercise Intervention**	**Effects on Tumor Progression**	**Ref.**
BC women	-CTRL: healthy women (*n* = 7) -BC: breast cancer patients receiving adjuvant chemotherapy (*n* = 20)	-CTRL: acute bout of ergometer cycling at 55% of VO_2_peak for 2 h -BC: 1 h resistance whole-body training and 30 min of high-intensity spinning (pulse > 80% HR max) on stationary ergometer bicycles	↓ cell viability by both CTRL and BC group exercise-conditioned sera: 19% and 11%, respectively, in MCF-7; and 9% and 13%, respectively, in MDA-MB-231 BC cells -45% of mice inoculated with MCF-7 cells pre-incubated with exercise-conditioned sera formed tumors	[221]
Subjects with stable spinal bone metastases undergoing radiotherapy	-Group A: resistance training (*n* = 30) -Group B: passive physical therapy (*n* = 30)	-Group A performed 30 min of 3 different exercises: in ‘‘all-fours’’ position, in the ‘‘gluteus arch’’ position, and in the ‘‘supine position’’ -Group B received 15 min of passive PA in form of breathing exercises	No differences in either overall or bone survival between groups A and B ↑ in local bone progression in 16.7% group B vs. A	[229]
Stage IIB–IIIC BC women	-AC group: combination with doxorubicin– cyclophosphamide (*n* = 10) -AC+AET group: aerobic exercise training (*n* = 10)	-AC received 4 cycles of doxorubicin (60 mg/m^2^) in combination with cyclophosphamide (600 mg/ m^2^) -AC+AET had also performed 12 weeks of 3 cycle ergometer sessions/week at 60–100% of VO2 peak, 30 to 45 min/session.	↓ IL-1β in both groups ↓ IL-2 in AC+AET group vs. AC ↑ IL-8 in AC+AET group vs. AC ↑ of 38% in tumor vascularization in AC+AET group vs. AC	[230]
Stage I–III CRC subjects	-CTRL: usual-care control (*n* = 13) -Low-dose EX: 150 min/week aerobic exercise (*n* = 14) -High-dose EX: 300 min/week of aerobic exercise (*n* = 12)	EX groups provided with an in-home treadmill and a HR monitor, performed 6 months aerobic training at 50–70% of the HR max	↓ circulating tumor cells in both low-dose and high-dose EX vs. CTRL ↓ BMI, insulin, and soluble ICAM-1 in all 3 groups	[231]
Stage II or III CRC subjects	-CTRL: CRC subject (*n* = 15) -EX group: CRC subject (*n* = 27)	EX group performed home-based 50 min circular workout 3 times/week, composed of a series of aerobic and anaerobic exercises	↑ Serum leptin in CTRL group ↓ Serum adiponectin in CTRL group ↑ Serum adiponectin in EX group	[244]
BC overweight or obese women	-EX group: (*n* = 37)	24 bi-weekly sessions of 25 min aerobic exercise using static bicycles at 70% of max workload and 25 min muscle strengthening exercises	↓ metabolic risk biomarkers and insulin resistance before vs. after training ↓ leptin, adiponectin, and BMI before vs. after training	[245]
Stage I–III BC obese women	-CTRL: BC obese women (*n* = 10) -EX group: BC obese women (*n* = 11)	16 weeks of 150 min/week of aerobic exercise, including treadmill walking/running, rowing machine, or stationary bicycle at 65–80% of HR max combined with 2–3 days of resistance exercise	↓ CRP, leptin, IL-6, IL-8 in EX group vs. CTRL ↑ adiponectin in EX group vs. CTRL ↓ macrophages M1 and ↑ M2 in EX group vs. CTRL ↓ AT secretion of IL-6 and TNFα ↑ IL-12 in EX group vs. CTRL	[246]

BC: Breast Cancer; BMI: Body Mass Index; CM: Culture Medium; CRC: Colorectal Cancer; CRP: C-Reactive Protein; CXCL1: chemokine (C-X-C motif) ligand 1; CTRL: control group; EMT: Epithelial–Mesenchymal Transition; EX: exercise group; FABP4: fatty-acid-binding protein 4; HR: heart rate; HUVEC: Human umbilical vein endothelial cells; HIF-1α: Hypoxia-Inducible Factor-1α; ICAM-1: Intercellular Adhesion Molecule 1; MCP-1: Monocyte chemoattractant protein-1; MNU: 1-methyl-1-nitrosourea; PDGF-BB: Platelet-Derived Growth Factor-BB; PPAR-γ: peroxisome proliferator-activated receptor gamma; TGF-β1: Transforming growth factor beta 1; VEGF: Vascular Endothelial Growth Factor.

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
