# Peer review of "Another Weapon against Cancer and Metastasis: Physical-Activity-Dependent Effects on Adiposity and Adipokines"

_ijms, 2021, doi:10.3390/ijms22042005_

Round 1

Reviewer 1 Report

For the manuscript ijms-1066798 „Another weapon against cancer and metastasis. Physical 2 activity-dependent effects on adiposity and adipokines”
The manuscript could have been interesting and uselful for clinicists and physiotherapists, but has been inadequately presented. When I was reading the manuscript, I got the feeling that it was written by two people and one of the did not know what the other author is writing about.
The manuscript requires significant improvement.

The manuscript is long, and as some points do not introduce any new information. There are also some substantive mistakes.

Comments:
Line 31. A sentence cannot start from a numer – please change it.
Line 39-40. Physical inactivity, or more appropriately the sedentary behaviour, represents a widespread cause of mortality....are you sure? In my opinion that is not true - please change it

Line 39-70 is a repetition of point 6

3. Direct effects of adipokines on tumour development, growth and metastasis
In my opinion, section 3 does not add any novelty to the manuscript and is redundant. The direct effects of adipokines on tumor development… is known. In the "Adipokines" section, eg 3.1. Adiponectin, 3.2. Leptin, 3.3. Appeal… 3.7. Chemerin… etc. There is not a single opinion on adipokines vs physical activity vs tumors I propose to change point 3, for example…. The influence of physical activity on adipokines in the development and course of cancer....
or by removing it, because in point 4 it partially explains the above issue. However, point 4 is chaotically described. In one sentence the authors describe research in mice, then in humans, and again mice, rats….

4. Effect of physical activity on adipose tissue, low-grade inflammation and tumour progression

Line;445_ Chronic exercise and physically active lifestyle are non-pharmacological interventions…. Chronic exercise? should be _ Regular exercise

Please change the entire section 4. eg. Line 576_ In general, chronic exercise determines

Does the Line- 578- 580_ following sentence applies to your conclusion or other researchers?

„However, it must always be kept in mind that an obese person does not have the same physical abilities as healthy persons and therefore cannot support exercise regimens with the same intensity and duration.” Please explain on the example of research, explain why an obese person does not have the same physical abilities as healthy people…

6. Practical recommendations and conclusive remarks

At least half of the manuscript is not afield. Please shorten the manscript and make sure it is pertinent. The manuscript requires significant improvement

Author Response

For the manuscript ijms-1066798 „Another weapon against cancer and metastasis. Physical 2 activity-dependent effects on adiposity and adipokines”
The manuscript could have been interesting and uselful for clinicists and physiotherapists, but has been inadequately presented. When I was reading the manuscript, I got the feeling that it was written by two people and one of the did not know what the other author is writing about.
The manuscript requires significant improvement.

The manuscript is long, and as some points do not introduce any new information. There are also some substantive mistakes.

The authors are grateful to this Reviewer for her/his comments and have spent efforts to improve the manuscript according to the precious recommendations. In particular, the manuscript have been revised in order to improve the form and shortened (by 5500 words) in its generalist parts. The authors hope that this new version will meet the Reviewer’s expectations.

Comments:
Line 31. A sentence cannot start from a numer – please change it.

The number has been turned into a written digit.

Line 39-40. Physical inactivity, or more appropriately the sedentary behaviour, represents a widespread cause of mortality....are you sure? In my opinion that is not true - please change it

The sentence has been changed in “Physical inactivity and sedentary behaviour represent widespread causes of mortality…”

Line 39-70 is a repetition of point 6

We agree with this Reviewer. Section 1 and section 6 have been reorganized in order to focus on the most important information, to avoid repetitions and to make the reading fluent.

  1. Direct effects of adipokines on tumour development, growth and metastasis
    In my opinion, section 3 does not add any novelty to the manuscript and is redundant. The direct effects of adipokines on tumor development… is known. In the "Adipokines" section, eg 3.1. Adiponectin, 3.2. Leptin, 3.3. Appeal… 3.7. Chemerin… etc. There is not a single opinion on adipokines vs physical activity vs tumors I propose to change point 3, for example…. The influence of physical activity on adipokines in the development and course of cancer....or by removing it, because in point 4 it partially explains the above issue. However, point 4 is chaotically described. In one sentence the authors describe research in mice, then in humans, and again mice, rats….

This concern have been considered in the hard revision of the manuscript. A linear trace has drawn. The title of the section has been modified according to the suggestion.

  1. Effect of physical activity on adipose tissue, low-grade inflammation and tumour progression

Line;445_ Chronic exercise and physically active lifestyle are non-pharmacological interventions…. Chronic exercise? should be _ Regular exercise

Please change the entire section 4. eg. Line 576_ In general, chronic exercise determines

Does the Line- 578-580_ following sentence applies to your conclusion or other researchers?

„However, it must always be kept in mind that an obese person does not have the same physical abilities as healthy persons and therefore cannot support exercise regimens with the same intensity and duration.” Please explain on the example of research, explain why an obese person does not have the same physical abilities as healthy people…

Even regarding this comment, the text have been thoroughly revised and only relevant information have been kept and discussed. The term chronic exercise has been modified into “regular exercise” all over the text according to this reviewer’s suggestion.

  1. Practical recommendations and conclusive remarks

At least half of the manuscript is not afield. Please shorten the manscript and make sure it is pertinent. The manuscript requires significant improvement

This section has been implemented with paragraphs taken from the introduction in order to increase readability

Reviewer 2 Report

In this article, the authors revise and discuss current evidences on the effects of physical activity on adipokines as mediators of metabolic inflammation, as well as the molecular mechanisms underlying adipokine-mediated impact of physical activity and body weight on tumour development, growth and metastasis. The approach is holistic and very interesting, representing a valuable contribution to the field. So congrats to the authors overall.

The introduction is well organized and comprehensively described. This section includes important concepts such as lifestyle, physical inactivity and sedentary behaviour. However, it would be advisable to review how these concepts are used throughout the document (please see also below).

The introduction analyzes the link between physical inactivity and overweight/obesity and risk of cancer and shows how necessary specific competences to prescribe exercise are in order to implement exercise programs in adult patients. In this regard, one of the main concerns in developed countries is how to address the malnutrition problems (too much or too little) in children and adolescents with cancer. Thus, it would be nice if the authors could elaborate more on the most relevant aspects related to this topic and on how exercise could help to improve the health status of these patients.  Childhood/adolescent cancer is indeed the great forgotten in general (vs adult tumors).

The following sections analyze in depth the consequences of sedentary behaviour and obesity in terms of cancer risk. The authors also describe the role of the adipose tissue as an endocrine organ and discuss how the immune cells located inside this tissue become polarized to type 1 or type 2 phenotypes depending on the health status of the adipocytes. However, the same idea is repeated many times (for example, paragraph beginning on line 700 +  line 704), and the scientific evidence that underpins this idea is explained without links between them. This aspect makes the reading difficult. For this reason, I think the authors should try to focus their ideas, and avoid repetitions and being too verbose at places. I think the same advice and comment applies to the whole Ms in general: please be more specific/concise and less verbose/repetitive. Also, please make sure you stick to the same concepts and definitions consistently throughout the Ms.

The authors state that: “The effects of exercise on tumour growth are well studied (line 712)”. In this regard, the authors should specify to which type of cancer and studies (in vitro, preclinical or clinical evidence) does this statement applies.

A ref is needed for the sentence “However, exercise has a positive effect on cancer in terms of volume reduction and tumour growth” (line 770).

Section number 4 (Effect of physical activity on adipose tissue, low-grade inflammation and tumour progression) deals with the benefits of exercise in cancer patients. Adding and discussing those studies analyzing the combined effect of exercise + a specific diet could improve the quality of this section.

Section number 5 does not clearly explain how the physical activity-dependent effect on adipokine and low-grade inflammation affects the risk of developing metastasis. A better explanation would be appreciated here. 

The practical recommendations shown in section number 6 are very useful for cancer patients in general. However, the incorporation of practical recommendations for patients with cancer + also obesity problems could further improve the quality of this section.

Figure 1. Are you representing the effect of physical activity or the effect of exercise? The title of this figure could be improved. Speaking of which, a clear definition and delineation of what is physical activity (PA) vs what is meant by physical exercise is needed. In general, PA applies to epidemiological evidence and physical exercise applies to preclinical/mechanistic research.

Table 2. I would like to know what type of inclusion criteria (for example, study design) was used to include a given paper in this table. Maybe the incorporation of papers studying the effect of exercise on metastasis would be nice. Please, revise abbreviations. 

My comments related to editorial/technical issues are listed below:

Much more consistency is needed with the use of abbreviations. Decreasing the number of  abbreviations could ease readability. I would stick to common (well-accepted) abbreviations in medical sciences (eg, BMI, CVD etc.)

Line 214 – please delete a coma in …CRC, ”,” endometrial…

Line 222 – please delete a space in …growth “ “, proliferation,…

Line 236 – please delete a space in …markers “ “, suggesting…

Line 238 – please delete “is a”

Line 247 – please delete a space in …stimulates “ “ inflammation…

Line 268 – please delete a space in …SW480 “ “, leptin,…

Line 317 – please delete a dot in …a “.” Mechanism…

Line 371 – please delete a space in …but “ “not…

Line 549 – “B” of breast in capital letter

Line 586 – please add a space in …5% “ “ tilt…

Line 657 – please delete space in … 75” “% VO2max…

Please look over the manuscript for these and other potential editorial/secretarial issues.

Author Response

In this article, the authors revise and discuss current evidences on the effects of physical activity on adipokines as mediators of metabolic inflammation, as well as the molecular mechanisms underlying adipokine-mediated impact of physical activity and body weight on tumour development, growth and metastasis. The approach is holistic and very interesting, representing a valuable contribution to the field. So congrats to the authors overall.

The authors are grateful to this Reviewer for her/his positive and encouraging comment. The authors are hopeful that this new version of the manuscript will meet the Reviewer’s expectations.

The introduction is well organized and comprehensively described. This section includes important concepts such as lifestyle, physical inactivity and sedentary behaviour. However, it would be advisable to review how these concepts are used throughout the document (please see also below). The introduction analyzes the link between physical inactivity and overweight/obesity and risk of cancer and shows how necessary specific competences to prescribe exercise are in order to implement exercise programs in adult patients. In this regard, one of the main concerns in developed countries is how to address the malnutrition problems (too much or too little) in children and adolescents with cancer. Thus, it would be nice if the authors could elaborate more on the most relevant aspects related to this topic and on how exercise could help to improve the health status of these patients. Childhood/adolescent cancer is indeed the great forgotten in general (vs. adult tumors).

The authors agree with this analysis. However, as known in children and adolescents tumours onset is not properly related to lifestyle, or at least to physical activity status. For sure, however, a physically inactive lifestyle, together with a malnutrition (hypo- or hypernutrition status) may predispose to a future development of tumours. Since the request of reducing the text, the authors have added a sentence in the conclusion section related to this aspect (last sentence).

The following sections analyze in depth the consequences of sedentary behaviour and obesity in terms of cancer risk. The authors also describe the role of the adipose tissue as an endocrine organ and discuss how the immune cells located inside this tissue become polarized to type 1 or type 2 phenotypes depending on the health status of the adipocytes. However, the same idea is repeated many times (for example, paragraph beginning on line 700 + line 704), and the scientific evidence that underpins this idea is explained without links between them. This aspect makes the reading difficult. For this reason, I think the authors should try to focus their ideas, and avoid repetitions and being too verbose at places. I think the same advice and comment applies to the whole Ms in general: please be more specific/concise and less verbose/repetitive. Also, please make sure you stick to the same concepts and definitions consistently throughout the Ms.

The whole manuscript has been revised and considerably shortened. The authors hope that this new version is clearer, but for sure less verbose, and more focused than the previous.

The authors state that: “The effects of exercise on tumour growth are well studied (line 712)”. In this regard, the authors should specify to which type of cancer and studies (in vitro, preclinical or clinical evidence) does this statement applies.

The section has been revised. These aspects have been better underlined.

A ref is needed for the sentence “However, exercise has a positive effect on cancer in terms of volume reduction and tumour growth” (line 770).

The following reference has been added in support of the sentence 10.1016/j.cmet.2017.09.015

Section number 4 (Effect of physical activity on adipose tissue, low-grade inflammation and tumour progression) deals with the benefits of exercise in cancer patients. Adding and discussing those studies analyzing the combined effect of exercise + a specific diet could improve the quality of this section.

The authors agree with this comment. Where available this information has been implemented. However, as reported at the end of section 4, this aspect has been almost neglected.

Section number 5 does not clearly explain how the physical activity-dependent effect on adipokine and low-grade inflammation affects the risk of developing metastasis. A better explanation would be appreciated here.

The section has been revised. These aspects have been better underlined.

The practical recommendations shown in section number 6 are very useful for cancer patients in general. However, the incorporation of practical recommendations for patients with cancer + also obesity problems could further improve the quality of this section.

The section has been revised, including aspects previously reported in the introduction.

Figure 1. Are you representing the effect of physical activity or the effect of exercise? The title of this figure could be improved. Speaking of which, a clear definition and delineation of what is physical activity (PA) vs. what is meant by physical exercise is needed. In general, PA applies to epidemiological evidence and physical exercise applies to preclinical/mechanistic research.

The term “exercise oncology” has been replaced by “physical activity. The legend has been modified to make it more clear.”

Table 2. I would like to know what type of inclusion criteria (for example, study design) was used to include a given paper in this table. Maybe the incorporation of papers studying the effect of exercise on metastasis would be nice. Please, revise abbreviations.

As reported in the text, studies about the effect of PA intervention on metastasis risk and recurrence are limited. Also according to the request of shortening the text, the authors think that adding additional information about the included study may generate further confusion. The authors intended the discussion of these papers in discursive terms, just to talk about the main evidences available. The aim of the table is, then, to summarize these information and to shed a light on the difficult comparison among these studies.

My comments related to editorial/technical issues are listed below:

Much more consistency is needed with the use of abbreviations. Decreasing the number of abbreviations could ease readability. I would stick to common (well-accepted) abbreviations in medical sciences (eg, BMI, CVD etc.)

The authors agree with this reviewer about the need to simplify the manuscript.

Where possible the explanation of the acronyms has been avoided. However, the authors mainly stayed at the editorial rules that ask that specification at first mention.

Line 214 – please delete a coma in …CRC, ”,” endometrial…

Line 222 – please delete a space in …growth “ “, proliferation,…

Line 236 – please delete a space in …markers “ “, suggesting…

Line 238 – please delete “is a”

Line 247 – please delete a space in …stimulates “ “ inflammation…

Line 268 – please delete a space in …SW480 “ “, leptin,…

Line 317 – please delete a dot in …a “.” Mechanism…

Line 371 – please delete a space in …but “ “not…

Line 549 – “B” of breast in capital letter

Line 586 – please add a space in …5% “ “ tilt…

Line 657 – please delete space in … 75” “% VO2max…

Please look over the manuscript for these and other potential editorial/secretarial issues.

Where still present, in the current version of the manuscript, these typos have been amended

Round 2

Reviewer 1 Report

Dear authors - good job, thank you!

Author Response

Dear Reviewer,

thank you for effort in making this manuscript suitable for publication.

Sincerely,

Veronica Sansoni

Reviewer 2 Report

Thank you so much for taking under consideration my comments and suggestions. I would like to extend my congratulations to the authors for this paper, which represents a valuable contribution to the field.  

Author Response

Dear Reviewer,

thank you for effort in making this manuscript suitable for publication.

Sincerely,

Veronica Sansoni

This manuscript is a resubmission of an earlier submission. The following is a list of the peer review reports and author responses from that submission.

Round 1

Reviewer 1 Report

General comment

The topic is very interesting in the exercise oncology and molecular science fields, where the manuscript ties physical activity and adipocyte induced molecular factors. However, the authors should carefully consider the choice of information to strengthen the evidence that the authors are trying to establish through the manuscript. Otherwise, this manuscript is only presenting the information. Furthermore, it will be appreciated if the manuscript presents more evidence if there is any, for each adipokine in response to exercise. It will help understand the connection between the exercise/increased physical activity and cancer in terms of adipokines. Lastly, the manuscript requires extensive editing to greatly improve English grammar and expression. As there are so many issues regarding the content of the manuscript, I have not provided any specific comments on this.

Detailed comment

  • For the introduction the authors tried to develop the importance of physical activity in the progression of cancer and metastasis. However, some of the information presented seems to be not necessary and requires more clarity as to the connections.

  1. I can not find the relevance of the first paragraph with the manuscript. Moreover, why is it important to distinguish the levels of physical inactivity in the topic? The authors did not make any distinction in adipokine secretion and production by physical inactivity levels. I have similar comments for section 2. Is there any evidence presenting a difference of adipokine secretion or production in ‘sedentary’ and ‘inactive’? If not, is making this distinction in the introduction necessary?

  1. All the epidemiological data in the second paragraph makes it hard to focus on the main topic of the introduction; the connection between obesity (and obesity-related factors) and cancer are weakly presented and blunt the focus of the main topic – adipokine and cancer metastasis.

  1. Although the fourth paragraph suggests critical information for the principle of exercise prescriptions and recommendations, it seems it does not fit into the introduction section. I would suggest present later in section Practical recommendations and conclusive remarks.
  • In section Sedentary behavior, adiposity, low-grade inflammation: factor risk for tumor development and metastasis, the authors mention the distinction between sedentary behavior and physical inactivity; however, authors have not made a clear distinction between sedentary behaviors and physical inactivity in terms of adipokine secretion or expression. What are the connections between the difference in physical activity levels and adipokines? Is it only adipose tissue mass? If authors are not clearly explaining the difference between these two conditions related to adipokines, the distinction of physical inactivity levels will not strengthen the manuscript, only causes confusion.

  • The authors comprehensively explained the general roles, paracrine/autocrine effects, and endocrine effects for cancer. However, the authors presented limited information about the alteration in secretion or production of each adipokine in response to exercise/increased physical activity. I understand the authors tried to relate alteration of adipokine production/secretion with exercise by presenting the studies of IL-6, leptin, and adiponectin in the following section. However, it would strengthen the point that the authors are trying to make through the manuscript if they also present evidence of exercise responses for other adipokines as well. If there is not enough data for the alteration of each adipokine with exercise, please state in the manuscript, it would be helpful for readers to understand the gap between exercise/physical activity-induced adipokines and cancer and current stages of research.

  • The explanation of a few studies in the manuscript is far more detailed compared to others. It would be helpful for the readers to understand the contents if the authors present the same levels of information for each study. I understand the authors tried to highlight the important studies; however, some of the explanations contain detailed and complex information that would not be easy to understand for non-exercise scientist readers and overly emphasizes the protocol, not the results of the study.

For example, on page 19, it seems that study protocol is overly emphasized, not the results and implications that such a study provides:

“Wang et al. have focused on the interaction between platelets and cancer cells that is known to be involved in thrombosis and metastasis formation. 30 healthy sedentary men (average BMI 25.4kg/m2) underwent to either high intensity exercise (HIE) and mild intensity exercise (MIE) on a bicycle ergometer. HIE protocol started with 2 min of unloaded pedaling followed by a workload that was incremented by 20-30 W every 3 mine until subject’s exhaustion and after reaching the VO2 peak. The MIE protocal consisted, instead, of 40 min pedaling at 60% VO2 max.”

Given that, this narrative review and focusing on adipose tissue induced molecular players, the focus should be on the molecular results, not the exercise protocols. This comment applies to all the studies presented in section 5. How the PA-dependent effect on adipokine and low-grade inflammation affects the risk of developing metastasis.  

  • In section 1 in vitro studies, 5.2 In vitro/animal model, and section 5.3 Clinical studies, the authors have presented significant evidence for tumor regression via exercise. However, I could not find the relevance in the few studies presented in this section in terms of adipokine influences. In fact, these studies only present the effect of exercise in cancer, with just a few studies showing the involvement of exercise-induced alteration of IL-6, adiponectin, and leptin in cancer. It would be far more informative if the authors present more evidence that shows adipokine involvement in restricting tumor metastasis via exercise stimulus.  

For example, in section 5.1 In vitro studies and section 5.3 Clinical studies, although these studies explain one of the mechanisms for reducing cancer metastasis via exercise-induced shear stress, I cannot find the relevance in this manuscript, where the authors are discussing the effects of adipokines in cancer.

- Section 5.1 In vitro studies: “In a microfluidic system, it has been demonstrated that higher shear stress comparable to an activity of intense physical exercise is able to kill circulating cancer cells. the microfluidic system mimic the blood pulshing circulation with higher shear stress (60 dynes/cm2) being comparable to one generated within the femoral artery during intensive exercise. Different cancer cells (MDA-MB-231, UACC-893 from BC, A549 from lung, and 2008 from the ovary) are pumped into the system and subjected to different shear stress. 4h of higher shear stress kill 90% of circulating tumour cells, that were killed in two way: necrosis during the circulation and apoptosis within 24h post circulation. The mechanism by which cells are killed probably lies in damage to the cytoskeleton.”

- Section 5.3 Clinical studies: “In a randomized clinical trial 23 stage I-III CRC patients underwent for 6 months to either a low-dose (150 min per week) or high-dose (300 min per week), aerobic activity by means of an in-house treadmill, at 50–70% of the age-predicted maximum HR. Results showed a significant reduction, compared to baseline, in circulating tumour cells, microfluidically detected throughout antibodies against the epithelial cell-adhesion molecules.”

In section 6. Practical recommendation and conclusive remarks, I was not able to locate the practical recommendation concerning the ‘fat’ in cancer patients. As a reader, I would expect that adipose tissue-induced adipokines may have a regulatory role in cancer. Although authors made the general recommendations for cancer, some thoughts or rationales in this section for exercise recommendations related to adipokines in cancer patients are needed. I do understand there is limited understanding in exercise-dependent regulation of adipokines, but it should not close the section by saying only adiponectin is a promising therapeutic factor.